# Problematic Social Media Use and Cyber Aggression in Italian Adolescents: The Remarkable Role of Social Support

**DOI:** 10.3390/ijerph19159763

**Published:** 2022-08-08

**Authors:** Alberto Borraccino, Noemi Marengo, Paola Dalmasso, Claudia Marino, Silvia Ciardullo, Paola Nardone, Patrizia Lemma

**Affiliations:** 1Department of Public Health and Paediatrics, University of Torino, 10126 Torino, Italy; 2Department of Developmental and Social Psychology, University of Padova, 35131 Padova, Italy; 3National Centre for Disease Prevention and Health Promotion, Italian National Institute of Health, 00161 Rome, Italy

**Keywords:** cyber aggression, problematic social media use, social support, adolescents’ health, HBSC

## Abstract

The recent increase in electronic and social media use among young people has highlighted the importance of focusing on problematic social media use (PSMU) and the concurrent phenomenon of cyber aggression, as well as the role of social support. As part of the 2018 Health Behavior in School-aged Children study in Italy, this study aims to explore the role of family, peer, and teacher support in the association between cyberbullying and PSMU. Methods: Data were collected from 4183 school classes in Italy for a total of 58,976 adolescents aged 11, 13, and 15 years. The prevalence of cyber aggression (both cybervictimization and perpetration) and PSMU was estimated across subgroups of different age, gender, and geographical residence. A set of multivariable logistic regressions was then used to investigate the association between cyberbullying and PSMU, considering the effect of social support. Results: Cybervictimization was more frequent among girls than in boys. PSMU was higher in 11-year-old boys and 13-year-old girls; 8.3% and 12.7%, respectively. Social support was highest in 11-year-olds, for both sexes, and then decreased with increasing age. The risk of cyberbullying, both suffered and perpetrated, was higher in girls and in the presence of PSMU. Social support showed to be highly protective against PSMU and cyberbullying in all ages and both genders. Conclusion: Although cyber aggression is less represented in Italy than in other European countries, it is likely to increase. PSMU appears to be an important contributor to the risk of cyber aggression; however, social support has been shown to be capable of reducing the risk of both phenomena. Public health policies fostering familiar and school support can help protect adolescents’ mental health, reducing the risk of problematic media use and cyberbullying.

## 1. Introduction

Cyber aggression and problematic social media use (PSMU) negatively affect adolescents’ health, especially in girls, who are more often victims of cyberbullying [1,2,3,4]. Excessive and maladaptive use of social media (including social networking sites and instant messaging apps), combined with emotional instability, extroversion, and lack of awareness, can exacerbate the phenomena [3,5,6]. According to recent literature, cyber aggression has been defined as an intentional act aimed at harming another person or persons through computers, mobile phones and other electronic devices and, regardless of its recurrence—shifting to an act of cyber bullying—it is perceived negatively by the victim [7]. The highest risk of cybervictimization was reported to occur between 12 and 15 years old [1,3] and was further shown to be associated with PSMU [4,8]. As for the latter, unlike the agreement of what cyber aggression and cyberbullying are, the interest in PSMU among adolescents has gradually increased over the last decade. Although there is still no shared understanding of what PSMU is and how it should be measured, it has been linked to a greater likelihood of depressive symptoms, health complaints, anxiety, lower life satisfaction, and social isolation [4,8,9,10]. Therefore, given the potential damage to adolescent mental and social health, it should become a public health priority to deepen knowledge of both PSMU and cyber aggression, and to identify any moderators capable of mitigating the occurrence of such harmful combinations. Among the likely known factors, social support, in its various components (school, family, or peer support), has already shown its favorable effects [11,12,13]. Indeed, the environment in which adolescents grow up, combined with specific personality characteristics, can be a positive determinant of adolescent health. Moreover, social support has been shown to have a positive role in an individual’s wellbeing. When young people feel supported in their daily choices, they become less vulnerable to those stressful events that can foster behaviors potentially harmful to their health [14,15,16].

School represents the primary social environment for adolescents. Improving collaboration with teachers to ensure optimal school support becomes critical for reducing PSMU and preventing cyberbullying. Unfortunately, the existing scarcity of evidence on PSMU and internet addiction prevention programs does not offer opportunities to provide evidence-based guidance to schools, and makes it difficult to evaluate the initiatives currently in place [17]. Additionally, school and family can act synergistically in supporting young people, as coziness and perceived protection from family has proven to be protective against cyberbullying, increasing interest in the role of social support [11]. Moreover, school can intervene in situations where family support is poor or completely lacking [14,18]. In addition to school and family, peer support has also been shown to reduce anxiety and depression in youth, leading to a reduced risk of cybervictimization [1].

Studies on the contributory role of social support against aggressive behaviors in the teenage years are of increasing interest in international literature, but research on Italian adolescents is still poor. Therefore, this study aims to explore the role of family, peers, and teachers, supporting the association between cyber aggression (acted or suffered) and PSMU in Italy.

## 2. Materials and Methods

### 2.1. Study Population

This study is based on the Italian Health Behaviors in School-aged Children (HBSC) 2018 study. HBSC is a World Health Organization (WHO) Collaborative Cross-National Survey run every four years. In 2018, it included 50 different countries across and outside Europe. The sampling procedure and study methodology followed regularly updated international research guidelines [19].

As for the Italian Country, the Ministry of Health and the Ministry of Education have adopted HBSC as the national referral surveillance for adolescents. Therefore, the national sample was increased to reach regional representativeness [20]. According to the international protocol, the school class was the primary sampling unit, and the participating schools were identified via systematic sampling from the Ministry of Education, University and Research list of all public and private schools. The final analytical sample included 4183 classes comprising 58,976 children aged 11, 13, and 15 years old. The response rate was 86.3% [20].

### 2.2. Measures

Cyberbullying victimization and perpetration: items on cyberbullying were introduced in the 2014 HBSC survey, using an adapted version of the bullying victimization questionnaire [21]. Students were asked how often, in the past two months, they had perpetrated or had been the victims of cyberbullying. Response options were: “*I have not cyberbullied/been cyberbullied*”, “*once or twice*”, “*2 or 3 times a month*”, “*about once a week*”, “*several times a week*”, further dichotomized into “*never*” vs. “*once or more*” [4].

Problematic social media use (PSMU): PSMU was investigated through a nine-item, yes/no scale, validated as the Social Media Disorder Scale [22]. Respondents reported whether, in the past year, they: regularly could not think of anything else but social media (preoccupation), regularly felt dissatisfied because they wanted to spend more time on social media (tolerance), often felt bad when they could not use their social media (withdrawal), failed to spend less time on social media (persistence), regularly neglected other activities because of using social media (displacement), regularly had arguments with others because of their social media use (problem), regularly lied to parents or friends about the time spent on social media (deception), often used social media to escape from negative feelings (escape), and had serious conflicts with parents or siblings because of their social media use (conflict) [9]. Internal items consistency was esteemed through the tetrachoric correlation matrix, reporting an alpha of 0.87 [23]. Participants with three or more missing values were excluded. PSMU was defined if participants answered “yes” to 6 or more items [22].

School support: This was measured using the teacher and the classmate support scale. The scale includes three items for teachers (care—“*I feel that my teachers care about me as a person*”, acceptance—“*I feel that my teachers accept me as I am*”, and trust—“*I feel a lot of trust in my teachers*”) and three items for classmates (cohesion—“*The students in my class enjoy being together*”, kindness—“*Most of the students in my class are kind and helpful*”, and acceptance—“*Other students accept me as I am*”). All responses were given on a 5-point Likert scale (from 0—“*strongly disagree*” to 4—“*strongly agree*”). To get the school support score, results for teacher and classmate scales were summed individually and each sum was divided by three; consequently, the two results were summed and divided by two. High school support was defined as a score ≥2.5, and low school support as a score <2.5 [24]. Cronbach’s alpha for teacher, classmate support, and school support were 0.75, 0.73, and 0.72, respectively.

Family support and peer support: Family support and peer support were assessed through a four-item validated scale [25]. Family support items regarded supportiveness (“*My family really try to help me*”), emotional aid (“*I get the emotional help and support I need from my family*”), confidence (“*I can talk about my problems to my family*”), and encouragement (“*My family is willing to help me make decisions*”). Items for peer support were about receiving help (“*My friends really try to help me*”), aid (“*I can count on my friends when things go wrong*”), affinity (“*I have friends with whom I can share my joys and sorrows*”), and confidence (“*I can talk about my problems with my friends*”). Responses in both scales were given on a 7-point Likert scale (from 0—“*very strongly disagree*” to 6—“*very strongly agree*”). Participants with one or more missing items were excluded. Scores were summed and then divided by four, and high support was defined for a score equal to or higher than 5.5, on each scale [26]. Cronbach’s alpha for family and peer support was 0.90.

Family Affluence Scale (FAS): FAS is a validated, six-item scale used to assess adolescents’ socioeconomic status through information on family ownership (number of computers owned, whether the family has a car and a dishwasher, whether the adolescent reports having their own bedroom, number of bathrooms in the house, and holidays taken in the past year) [27]. In concurrence with other published studies, the FAS score was categorized as low (≤6), medium (7–9), and high (≥10) [28,29].

### 2.3. Statistical Analysis

Descriptive analyses, by age and gender, were used to assess the prevalence of cyberbullying victimization and perpetration, PSMU, and the perception of having high social support (school, family, and peers) among Italian adolescents. A set of logistic regression analyses were performed to study the effect of high social support in moderating the association between cyberbullying victimization or perpetration. Model 1 assessed the effect of gender in the three age categories for cyberbullying victimization and perpetration and for PSMU. Model 2, only for cyberbullying, added the independent variable PSMU and evaluated its effect with respect to model 1. Model 3 added the moderating effect of social support, in its three components, to cyberbullying and PSMU. Results were reported as Odds Ratio (OR) and relative 95% Confidence Interval (95% CI). All analyses were performed using STATA software 17 (Stata Corp LP, College Station, TX, USA) and a two-tailed *p* value < 0.05 was considered significant.

## 3. Results

Table 1 reports the prevalence of all the variables considered in the analysis, stratified by age and gender. The distribution of cyberbullying victimization, cyberbullying perpetration, PSMU and social support prevalence, divided by Region, were reported in Appendix A. Cybervictimization was more frequent among girls than boys, with a peak in 11-year-old (10.6%) and 13-year-old (10.8%) girls. On the other hand, cyberbullying perpetration was more frequent in 13-year-old girls (7.4%) and 15-year-old boys (8.3%). PSMU was higher among the youngest: 8.3% in 11-year-old boys and 12.7% in 13-year-old girls, with notable differences between regions (see Appendix A). The perception of having high support (school, peers, and family) decreased with age in both genders; the highest values were observed in 11-year-old girls and boys for family support.

Table 2 reports the association between cyberbullying victimization and gender, PSMU, and social support through the three models implemented. The risk of cybervictimization was significantly higher among girls of all ages (model 1), with the lower risk in 11-year-old (OR 1.28, 95% CI 1.2–1.4) and the highest in 13-year-old (OR 1.68, 95% CI 1.5–1.9) girls. Model 2 showed a significantly high risk for PSMU for victims of cyberbullying of all ages.

The OR for PSMU was the highest in the 13-year-old age group (OR 2.74, 95% CI 2.4–3.1). When introducing the role of social support (Model 3), the risk of being cyberbullied reported a significant protective effect in all ages and for all sources of support, independently of PSMU, which again showed the highest risks, especially in 13-year-old girls (OR 2.43, 95% CI 2.1–2.8).

The association between cyberbullying perpetration, gender, PSMU, and social support was reported in Table 3. Model 1 showed that girls reported a significantly lower risk for cyberbullying perpetration in the 11-year-old and in the 15-year-old group (OR 0.84, 95% CI 0.7–0.9 and OR 0.81 95% CI 0.7–0.9, respectively).

Similarly, Model 2 PSMU showed a positive and significant association with cyberbullying perpetration in all age groups, which was the highest in the 13-year-old group (OR 3.18, 95% CI 2.8–3.7). Looking at Model 3, social support reported a significant protective effect on cyberbullying perpetration in all age groups. The only non-significant value observed was that of 13-year-old group for peer support (OR 0.95, 95% CI 0.8–1.1).

Table 4 reported the association between PSMU and gender (Model 1) and social support (Model 3). The risk of reporting PSMU among the three age categories was significantly higher in 13-year-old girls (OR 1.88, 95% CI 1.7–2.1) and 15-year-old girls (OR 1.83, 95% CI 1.6–2.1). School support and family support (Model 3) showed a significantly lower risk of PSMU in all age categories, while peer support reported a non-significant association.

## 4. Discussion

As reported for other European countries, in Italy, cyber aggression and cyberbullying represent an emerging phenomena among adolescents; cybervictimization is more frequent than cyberbullying perpetration and, in line with other studies, its occurrence decreases with age, although it is higher in girls, particularly at younger ages [1,2,3,4,30]. Cyberbullying perpetration showed a slight increase with age in both genders. The novelty of the approach was analyzing the occurrence of cyber aggression in the presence of PSMU and through the effect of perceived high social support.

Excessive social media use is considered an emerging problem: it is gaining increasing attention in the international scene as itself and for the mental health problems it can cause [6,30], but also for its significant association with cyberbullying [6,9,31]. Consistently, our study showed how PSMU represents a widespread risk factor for cyber aggression in adolescents [2,6,31]. It has, in fact, recently emerged as an important question to be tackled in Italy, placing our country in fourth place among European countries with the most intense use of social media [9].

Recent studies showed that girls tend to engage more frequently in online social interactions than boys of the same age; furthermore, they suffer more from the consequences of excessive use [2,32], experiencing depression, symptoms of psychosomatic origin and other forms of mental illness [5,33], as well as addictive behaviors [34]. Moreover, psychological stress and dissatisfaction with one’s life were shown to be higher in students who have integration problems at school. Compared to those who are appreciated by their peers, adolescents who feel rejected or disregarded are more likely to engage in passive-aggressive behavior, including cyber aggression. The odds are higher in girls than in boys [35].

Social support, consistently with other studies, has been reported to be a strong positive mediator of wellbeing and life satisfaction [14,15], a measure of protection against cybervictimization [36,37], and a positive moderator of the interaction between cyberbullying and PSMU. As a consequence, support coming from significant adult figures in schools [16,36] and from families [10,37] should be addressed as a key component to promote the psychophysical wellbeing of adolescents, and not only to reduce the risk of cyber aggression and PSMU [16,18].

The perception of having high peer support, on the other hand, was shown to be relatively less important in moderating the risk of cyberbullying and PSMU. However, our results confirmed that peer support can be an additional protective element against cybervictimization [1]. It has to be emphasized that peer support, relying mainly on popularity, may also represent a threat, pushing adolescents towards risky behaviors, as revealed via PSMU, or making them feel safe perpetrating cyberbullying [38,39]. In the latter case, family support does not seem to be successful in modifying the negative effects of peer support that encourages cyberbullying [37].

Our results revealed the need to further investigate the association between cyberbullying and maladaptive use of the internet, videogaming and streaming, and not only of social media. Indeed, the importance of early interventions to prevent problematic use of digital technologies (e.g., internet and social media) during adolescence is crucial to prevent negative mental health effects in late adolescence and adulthood, as they increase the risk of low self-esteem, depression, or suicidal thoughts [40].

Although not conclusively proven, based on our results and in accordance with other recent studies, it can be inferred that more appropriate use of social media could lead to a reduction in cyberbullying and a better quality of life among adolescents, especially in the developmental age group [41,42].

Given all the above, there is an urgent need for public health policies to strengthen those actions that, through the support of schools and families, may positively impact young people’s mental health and reduce cyber aggression. Given the pervasiveness of these phenomena, there is an increased need for social support policies, acting at a population level and not only targeting those at risk [35,43,44]. Moreover, prevention programs in schools focus on providing individual protective emotional skills, especially considering those positive psychological variables that seem to influence the behavior of adolescents [17], neglecting to act on social support, that has shown to exert an important protective role. To increase their effectiveness over time, while also adapting to changing environments, there is an urgent need for specific actions able to consider the cultural context, favoring participation, without hindering usage and dissemination of the emerging technologies.

Studying the role of social support in its various facets has become crucial for planning comprehensive educational programs for the health of adolescents. Indeed, observing the relationship of adolescents with their peers in the school environment reveals significant gender and age differences. For this reason, intervention programs should focus on the school environment, to both prevent situations of psychological distress and promote positive relationships between adolescents and their teachers and peers, aiming to enhance support and involve families in the process [35].

The above results should be considered in light of the study’s limitations and strengths. Firstly, the HBSC study, by its methodology, is a cross-sectional study, and therefore does not allow any conclusions on causality. However, the relationship between cyberbullying and PSMU was confirmed in longitudinal studies, and the same can be said for the moderating role of social support [6,39]. Regarding the latter, the dichotomization of the different variables to study social support could have led to a loss of information. Nevertheless, we have decided to faithfully adhere to the HBSC protocol recommendations [19] and to recent studies, so as to allow cross comparability [26]. The main strength of this study was the use of a large and representative Italian sample to investigate the association between cyberbullying, PSMU and social support.

## 5. Conclusions

Cyberbullying and PSMU are emerging problems among adolescents, and they are interrelated in Italy, as in other European countries. The problem is prevalent in young people and especially in girls, who are also more sensitive to negative consequences. The fact that our study confirms social support as a key element of prevention is of decisive importance for public health in order to intervene, at an early stage, in the possible further growth of cyberbullying and PSMU among adolescents.

## Figures and Tables

**Table 1 ijerph-19-09763-t001:** Prevalence of cyberbullying victimization, perpetration, Problematic Social Media Use (PSMU), and perception of high social support in Italy by age and gender (Italian HBSC 2018).

	11 y.o. Boys		11 y.o. Girls		13 y.o. Boys		13 y.o. Girls		15 y.o. Boys		15 y.o. Girls		Overall	
	N	%	N	%	N	%	N	%	N	%	N	%	N	%
cb victimization	842	8.6	1002	10.6	691	6.7	1078	10.8	545	5.8	739	7.8	**4897**	**8.4**
cb perpetration	653	6.7	538	5.7	739	7.1	743	7.4	774	8.3	619	6.6	**4066**	**7.0**
PSMU	736	8.3	759	8.6	721	7.2	1242	12.7	526	5.9	938	10.2	**4922**	**8.8**
high school support	8142	85.4	8029	87.0	7901	77.6	7083	71.9	6323	68.8	5617	60.1	**43,095**	**75.2**
high peer support	5997	62.1	6992	74.8	6138	60.3	7153	72.0	5504	59.7	6503	69.1	**38,287**	**66.3**
high family support	8128	81.8	7826	81.8	7872	75.2	7200	71.4	6493	69.0	6063	63.8	**43,582**	**73.9**
**Total**	**9940**		**9564**		**10,468**		**10,086**		**9412**		**9506**		**58,976**	

Missing in in boys and girls: cb victimization (*n* = 99, 1.1% and *n* = 80, 0.8% respectively); cb perpetration (*n* = 108, 1.2% and *n* = 71, 0.7%); PSMU (*n* = 443, 4.7%; *n* = 300, 3.2%); high school support (*n* = 222, 2.4%; *n* = 160, 1.7%); high peer support (*n* = 194, 2.1%; *n* = 100, 1.1%); high family support, no missing values.

**Table 2 ijerph-19-09763-t002:** Odds ratio (OR) and 95% confidence interval (CI) for cyberbullying victimization, by age, gender, PSMU, and social support (school peers and family); Italian HBSC 2018.

	CB Victimization	Model 1 ^a^		Model 2 ^a^		Model 3 ^a^	
		OR	95% CI	OR	95% CI	OR	95% CI
11 years	Girls	**1.28**	**(1.16–1.41)**	**1.29**	**(1.16–1.43)**	**1.38**	**(1.23–1.55)**
	PSMU	-	-	**2.44**	**(2.11–2.82)**	**2.10**	**(1.78–2.47)**
	High school support	-	-	-	-	**0.57**	**(0.49–0.65)**
	High peer support	-	-	-	-	**0.64**	**(0.57–0.73)**
	High family support	-	-	-	-	**0.56**	**(0.49–0.65)**
13 years	Girls	**1.68**	**(1.50–1.87)**	**1.55**	**(1.40–1.72)**	**1.57**	**(1.40–1.77)**
	PSMU	-	-	**2.74**	**(2.40–3.12)**	**2.43**	**(2.10–2.80)**
	High school support	-	-	-	-	**0.46**	**(0.41–0.52)**
	High peer support	-	-	-	-	**0.66**	**(0.59–0.75)**
	High family support	-	-	-	-	**0.64**	**(0.57–0.73)**
15 years	Girls	**1.44**	**(1.26–1.64)**	**1.38**	**(1.21–1.56)**	**1.37**	**(1.20–1.57)**
	PSMU	-	-	**2.33**	**(1.97–2.76)**	**2.02**	**(1.68–2.42)**
	High school support	-	-	-	-	**0.55**	**(0.48–0.63)**
	High peer support	-	-	-	-	**0.60**	**(0.52–0.68)**
	High family support	-	-	-	-	**0.57**	**(0.50–0.66)**

^a^ Model 1 is adjusted by FAS and region of residence; Model 2 adjusted for FAS, PSMU, and region of residence; Model 3 mutually adjusted for all variables; statistically significant results are reported in bold characters.

**Table 3 ijerph-19-09763-t003:** Odds ratio (OR) and 95% confidence interval (CI) for cyberbullying perpetration, by age, gender, PSMU, and social support (school peers and family); Italian HBSC 2018.

	CB Perpetration	Model 1 ^a^		Model 2 ^a^		Model 3 ^a^	
		OR	95% CI	OR	95% CI	OR	95% CI
11 years	Girls	**0.84**	**(0.74–0.95)**	**0.84**	**(0.74–0.96)**	0.89	(0.77–1.02)
	PSMU	-	-	**2.76**	**(2.33–3.27)**	**2.41**	**(2.00–2.91)**
	High school support	-	-	-	-	**0.58**	**(0.49–0.70)**
	High peer support	-	-	-	-	**0.85**	**(0.73–0.99)**
	High family support	-	-	-	-	**0.58**	**(0.49–0.68)**
13 years	Girls	1.05	(0.94–1.18)	0.94	(0.84–1.06)	0.91	(0.81–1.03)
	PSMU	-	-	**3.18**	**(2.76–3.66)**	**2.78**	**(2.39–3.23)**
	High school support	-	-	-	-	**0.62**	**(0.54–0.71)**
	High peer support	-	-	-	-	0.95	(0.84–1.09)
	High family support	-	-	-	-	**0.55**	**(0.48–0.67)**
15 years	Girls	**0.81**	**(0.71–0.91)**	**0.76**	**(0.67–0.86)**	**0.73**	**(0.64–0.84)**
	PSMU	-	-	**2.37**	**(2.00–2.80)**	**2.29**	**(1.91–2.73)**
	High school support	-	-	-	-	**0.72**	**(0.63–0.82)**
	High peer support	-	-	-	-	**0.86**	**(0.76–0.98)**
	High family support	-	-	-	-	**0.65**	**(0.57–0.74)**

^a^ Model 1 is adjusted by FAS and region of residence; Model 2 adjusted for FAS, PSMU, and region of residence; Model 3 mutually adjusted for all variables; statistically significant results are reported in bold characters.

**Table 4 ijerph-19-09763-t004:** Odds ratio (OR) and 95% confidence interval (CI) for PSMU, by age, gender and social support (school peers and family); Italian HBSC 2018.

	PSMU	Model 1 ^a^		Model 3 ^a^	
		OR	95% CI	OR	95% CI
11 years old	Girls	1.06	(0.95–1.18)	1.08	(0.97–1.21)
	High school support	-	-	**0.61**	**(0.53–0.71)**
	High peer support	-	-	1.03	(0.91–1.17)
	High family support	-	-	**0.54**	**(0.47–0.62)**
13 years old	Girls	**1.88**	**(1.70–2.07)**	**1.81**	**(1.63–2.00)**
	High school support	-	-	**0.68**	**(0.61–0.76)**
	High peer support	-	-	1.00	(0.90–1.12)
	High family support	-	-	**0.51**	**(0.46–0.57)**
15 years old	Girls	**1.83**	**(1.63–2.05)**	**1.76**	**(1.56–1.98)**
	High school support	-	-	**0.72**	**(0.64–0.81)**
	High peer support	-	-	0.92	(0.82–1.04)
	High family support	-	-	**0.60**	**(0.54–0.68)**

^a^ Model 1 is adjusted by FAS and region of residence; Model 3 mutually adjusted for all variables; statistically significant results are reported in bold characters.

## Data Availability

The data presented in this study are available in accordance with the Italian HBSC data access policy. Requests should be directed to paola.nardone@iss.it, a member of the National Centre for Disease Prevention and Health Promotion, Italian National Institute of Health.

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
