# Peer review of "Problematic Social Media Use and Cyber Aggression in Italian Adolescents: The Remarkable Role of Social Support"

_ijerph, 2022, doi:10.3390/ijerph19159763_

Round 1

Reviewer 1 Report

I would like to thank the Editor and the Authors for the opportunity to review the manuscript entitled: Problematic Social Media Use and Cyber Aggression in Italian Adolescents: The Remarkable Role of Social Support.

I find the contribution of relevant contents for the Journal and very clearly written. Therefore, I recommend this protocol to be published. Nonetheless, I think that the manuscript needs some minor revisions.

It would be important to make explicit the subtended theoretical model, that has guided the choice of the study variables, methods and analyses; this offers the key of data interpretation.

As concerns the applied statistics, could the Authors report test for interactions among the variables; it would be useful, also in consideration of their discussion.

Finally, the discussion largely repeats the results without in-depth discussion of the findings. In addition, the application of the findings to intervention needs to be fully explained. Some of the references that could be included to enrich both introduction and discussion could be:

Berte, D. Z., Mahamid, F. A., & Affouneh, S. (2021). Internet addiction and perceived self-efficacy among university students. International Journal of Mental Health and Addiction19(1), 162-176.

Throuvala, M. A., Griffiths, M. D., Rennoldson, M., & Kuss, D. J. (2019). School-based prevention for adolescent internet addiction: Prevention is the key. A systematic literature review. Current neuropharmacology17(6), 507-525.

Author Response

Reviewer #1’s Comments:

  1. It would be important to make explicit the subtended theoretical model, that has guided the choice of the study variables, methods and analyses; this offers the key of data interpretation.

Authors Reply (AR): To account for the reviewer suggestion we have partly enlarged the introduction, adding the needed references, by focusing on the role and the interest of the recent literature on social support.

  1. As concerns the applied statistics, could the Authors report test for interactions among the variables; it would be useful, also in consideration of their discussion.

AR: Interactions among variables, in particular: age against all the other variables and gender against all the other variables, were tested with the chi square (X2) and resulted all significant with a p<000.1

Due to collinearity only models 2 were run with the interaction for gender and PSMU. Age was also significant and for this reason and due to theoretical reasons, coherently with the shooll-age and developmental issues also reported in HBSC methodology report, models were run stratified by age.

To take into account the reviewer's suggestion, the adjustment for interaction, previously omitted because it was erroneously considered superfluous, has been reported in the footnotes.

  1. Finally, the discussion largely repeats the results without in-depth discussion of the findings. In addition, the application of the findings to intervention needs to be fully explained. Some of the references that could be included to enrich both introduction and discussion could be: Berte, D. et al (2021). International Journal of Mental Health and Addiction, 19(1), 162-176; and Throuvala, M. A., et al. (2019). Current neuropharmacology, 17(6), 507-525.

AR: We apologies for creating so much redundancy. We have chosen to discuss the results methodically, probably leaving too much repetition.

We thank the reviewer for having suggested two new references, which were included in the introduction and in the discussion coherently with their content. Please see the amendments on the marked manuscript copy.

The list of reference was finally enlarged to also include the suggestions from reviewer #2.

Reviewer 2 Report

Thank you for the opportunity to read and review the paper entitled „Problematic Social Media Use and Cyber Aggression in Italian Adolescents: The Remarkable Role of Social Support”. I think this topic is up-to-date. A lot of changes in the internet use have been seen particularly during the last couple of years, partly because of the Covid pandemic and partly because of recent advances in technology. In turn, switching to new technologies course some consequences such as overuse of social media, additions, cyberbullying, and many others. For this reason, I consider this paper an important piece of the literature.

The study is well written and designed with the large sample size being an advantage. The introduction well introduces the topic. Results are well described and illustrated with tables. The discussion puts the study well in the context of the available data. References seem to be recent and appropriately selected.

I have one question that would add value to your work. Do you have any examples of any public health policies and/or programs that would teach this young population how to appropriately use social media? And next, if those programs were successful in the reduction of problematic social media use among adolescent and diminishing the level of cyber aggression? Such information would give some practical insights into the topic.

Minor issues are with typos, particularly with improper use of hyphenation, punctuation and capitalization. I suggest that the manuscript is corrected by a proofreader or thoroughly read once again by the authors. Also, abbreviations should be spelt out at the first mention and then used as abbreviations ever after.

Author Response

Reviewer #2’s Comments:

  1. I have one question that would add value to your work. Do you have any examples of any public health policies and/or programs that would teach this young population how to appropriately use social media? And next, if those programs were successful in the reduction of problematic social media use among adolescent and diminishing the level of cyber aggression? Such information would give some practical insights into the topic.

AR: We thank the reviewer for the invitation provided. As also introduced in the discussion there I a scarcity of evidence of effectiveness of any prevention programmes acting on bullying and even less on PSMU (which is agreed to be a quite novel issue). This limited evidence does not offer the opportunities to provide the needed guidance to schools.

For what was in our competences and also following the reviewer #1’s comments we have enriched the cited literature, to provide a satisfactory reply to this comment. Please see the amendments on the marked manuscript copy.

  1. Minor issues are with typos, particularly with improper use of hyphenation, punctuation and capitalization. I suggest that the manuscript is corrected by a proofreader or thoroughly read once again by the authors. Also, abbreviations should be spelt out at the first mention and then used as abbreviations ever after.

AR: We thank the reviewer for the kind attention in revising the text. Manuscript was submitted to a proofreader as suggested and undergo a careful read by the authors. We hope the amendments were satisfactory.
